# Data-Driven Phenotyping from Foot-Mounted IMU Waveforms: Elucidating Phenotype-Specific Fall Mechanisms

**DOI:** 10.3390/s25247503

**Published:** 2025-12-10

**Authors:** Ryusei Sato, Takashi Watanabe

**Affiliations:** 1Graduate School of Engineering, Tohoku University, Sendai 980-8579, Miyagi, Japan; 2Graduate School of Biomedical Engineering, Tohoku University, Sendai 980-8579, Miyagi, Japan; t.watanabe@tohoku.ac.jp

**Keywords:** gait phenotyping, inertial sensor, fall risk, dynamic time warping, timed up and go test

## Abstract

A one-size-fits-all approach to fall risk assessment in older adults has critical limitations. This study aimed to overcome this by identifying distinct gait phenotypes and their specific fall mechanisms using foot-mounted IMU waveform data from 146 older adults (mean age 82.6 ± 6.2 years). A data-driven clustering algorithm identified four phenotypes (Robust, High-cadence, Intermediate, and Cautious), each with different fall prevalence rates (27–68%). Interpretable machine learning (SHAP) revealed that fall trajectories were phenotype-dependent. While physiological declines such as gait speed were the primary cause of falls in the Cautious group, fear of falling (FES-I) was the primary cause in the physically healthy Robust group, suggesting a psychological pathway. Consequently, the optimal Timed Up and Go (TUG) test screening cutoff varied across phenotypes, ranging from 11.95 s to 14.00 s, demonstrating the limitations of a one-size-fits-all approach. These findings demonstrate that fall mechanisms are phenotype-dependent, underscoring the necessity of a personalized assessment strategy to improve fall prevention.

## 1. Introduction

Older adults (aged ≥ 65 years) currently account for approximately 10% of the global population, and this percentage is expected to increase to 16% by 2050 [1]. Falls are a major public health issue for this demographic group, ranking as the second leading cause of unintentional injury-related death in those aged 60 and older. Annually, approximately 30% of individuals aged 65 and older experience at least one fall [2,3]. The consequences can be severe, including fractures and head trauma, which often lead to an increased need for long-term care and higher mortality rates. Therefore, assessing fall risk and implementing preventive strategies are critical public health priorities. Fall risk is multifactorial, with contributing factors including age-related physiological changes, muscle weakness, impaired balance, polypharmacy, multimorbidity, and psychological factors (e.g., fear of falling) [4]. However, comprehensive clinical evaluation of these risks is time-consuming and resource-intensive, highlighting the need for more efficient screening methods.

Traditional approaches to fall risk assessment in older adults commonly employ dichotomous classifications (e.g., high-risk vs. low-risk). This one-size-fits-all model, however, is increasingly challenged by the recognized heterogeneity of this population [5,6]. Even individuals labeled as fallers constitute a heterogeneous group with diverse gait patterns and risk profiles. The limitations of a single threshold screening method are exemplified by the Timed Up and Go (TUG) test [7,8,9]. While a cutoff of 13.50 s is often used to identify individuals at risk [10], its clinical utility is hampered by a low sensitivity of around 31%, meaning many at-risk individuals may be missed [11]. Therefore, there is a clear need for assessment strategies that move beyond simple binary classifications and incorporate multiple indicators to account for individual variability [7,8,9].

To address these issues, data-driven gait analysis approaches using clustering have attracted attention. By classifying older adults into highly homogeneous subgroups, clustering can potentially uncover hidden patterns within each population and extract features influencing fall risk with greater detail. A previous study reported that older adults could be classified into four groups with differing fall incidences (low risk, two intermediate-risk groups, and high risk) through unsupervised clustering based on multiple gait and balance metrics [12]. In that study, the actual fall incidence across clusters varied from 13% to 31%, and cluster-specific contributing factors such as reduced gait speed, poor balance, muscle weakness, presence of heart disease, and cognitive decline were identified. Another study demonstrated that clustering based on functional and biomechanical parameters during physical performance tests clearly divided older adults into three groups based on high and low physical performance, capturing population differences distinct from conventional classifications such as frail/non-frail or faller/non-faller [5]. In another assessment of physical function in older adults, a study used clustering of kinematic signals from the chair stand test to provide hidden assessment perspectives while maintaining an evaluation capability comparable or superior to traditional methods [13]. In our own previous study [14], we applied cluster analysis to the time-series data of acceleration and angular velocity in the sagittal plane obtained from the open dataset [15] measuring elderly individuals’ walking using foot-mounted inertial sensors. This process classified the gait patterns into distinct clusters, and the association between each cluster and fall history was then evaluated. Thus, clustering methods are useful as a new approach to shed light on the intra-group heterogeneity often overlooked by conventional dichotomous classifications in the assessment of gait and fall risk in older adults.

For more detailed gait assessment, the use of gait waveforms measured by accelerometers and gyroscopes has become increasingly common [16,17,18]. Approaches that compare the overall shape of waveforms have also gained attention. Methods like the Linear Fit Method (LFM) have been validated for their reliability and validity in assessing the similarity between the gait patterns of patients with cerebrovascular accident (CVA) and those of healthy individuals by independently quantifying the shape, amplitude, and offset of the waveform [16]. A significant challenge in accurate pattern comparison, such as waveform shape analysis, is accounting for the effects of velocity fluctuations and pattern stretching that occur between subjects and trials. An effective method for addressing this challenge is Dynamic Time Warping (DTW). In gait recognition research, DTW algorithm is used to calculate the distance between a reference gait stride and a test gait stride and has been reported to improve recognition performance [19,20]. This algorithm is useful for identifying an individual’s unique gait pattern and detecting gait changes due to disease, and we utilized it in our previous study to calculate the dissimilarity of raw data waveforms.

The purpose of this study was to elucidate the heterogeneity within the gait of older adults by analyzing the relationship of fall risk to a set of gait parameters and clinical test results for each gait phenotype. These phenotypes are derived from clustering the dissimilarity of raw gait waveform data, calculated using DTW. In this paper, the term Phenotype does not refer to a clinical diagnosis, but rather to a cluster of statistically similar movement patterns derived from IMU waveform data using a data-driven process. The specific objectives are as follows: 1—To demonstrate that the gait features contributing to the stratification of fall risk differ among phenotypes, by analyzing the feature importance of a Random Forest classification model. This suggests that the underlying mechanisms of falls vary across different phenotypes. 2—Using the TUG test as a clinical case study, to validate the appropriateness of a single-threshold screening approach across phenotypes with these different fall mechanisms. This includes not determining phenotype-specific optimal thresholds but evaluating the diagnostic trade-offs (e.g., sensitivity and precision) and clarifying the limitations of the one-size-fits-all paradigm.

Recent fall risk studies using IMUs have focused on using machine learning models (e.g., SVM, RF) to classify older adults into Fallers and Non-fallers. While these studies have succeeded in calculating a single risk score, they overlook the heterogeneity within the faller group. This approach fails to explain fundamental differences in underlying mechanisms among subjects. The objective of this study is fundamentally different from building conventional predictive models. We aim, first, to phenotypically classify elderly walking based on patterns inherent in the IMU waveform data itself, using a data-driven approach. Subsequently, using interpretable AI methods such as SHAP, we seek to elucidate how key biomechanical and psychological factors associated with falls differ across each phenotype. We propose that this phenotype-based approach is the key to transitioning from one-size-fits-all fall prevention measures to personalized prevention strategies.

## 2. Materials and Methods

The overall workflow of this study is illustrated in Figure 1. The process consists of seven steps, (A) to (G). (A) Data Input involves preprocessing the IMU waveforms contained in the elderly dataset. (B) Clustering method development involves setting the clustering method and conditions to be used in subsequent analysis. (C,D) Phenotyping involves clustering the elderly population using the identified method and dividing them into phenotypes with different characteristics. (E,F) Phenotype-specific Analysis involves analyzing fall history by phenotype to identify phenotype-specific fall mechanisms. (G) Clinical Validation involves verifying the limitations of the one-size-fits-all strategy by customizing the TUG threshold for each phenotype.

### 2.1. Dataset

In this study, we utilized the GSTRIDE database of older adults, which was published by García-Villamil et al. [15]. The GSTRIDE database contains health status assessment information for 146 older adults (mean age 82.6 ± 6.2 years), including both a group with a history of falls within the past year (fallers) and a group without (non-fallers). For each participant, in addition to demographic data and basic physical function assessment indices, various clinical test results for frailty and fall risk assessment have been collected and provided. Specific assessment items include results from functional tests such as the Global Deterioration Scale (GDS) [21], the five items of Fried’s frailty scale [22], the Short Falls Efficacy Scale-International (Short FES-I) [23], 4-meter gait speed, the Timed Up and Go (TUG) test, and the Short Physical Performance Battery (SPPB) [24]. Furthermore, this dataset provides raw acceleration and angular velocity data from an inertial measurement unit (IMU) on the foot during the participants’ walking tests, as well as gait parameters calculated from this data. The walking test involved each participant walking continuously for up to 30 min, both indoors and outdoors where possible, with measurements taken by an IMU attached to the dorsum of the foot. GSTRIDE database includes data from two systems: a custom IMU (104 Hz, with an iNEMO inertial module LSM6DSRX, STMicroelectronics) and a commercial IMU (128 Hz, Physilog 6 S, GaitUp). This study utilized data from the 146 participants recorded with the 104 Hz custom IMU to ensure signal consistency. The signals were pre-processed with a 4 Hz low-pass filter.

### 2.2. Features

In this study, we used gait-related indices, clinical test results, and anthropometric data including age and height, obtained from the aforementioned dataset, as explanatory variables (features). The features included the mean and standard deviation of the gait parameters derived from the IMU (Table 1). In this study, the gait data for each participant was segmented into individual strides.

### 2.3. Clustering Method Development

The development of the clustering algorithm included stages of data pre-processing, feature extraction, application of clustering algorithms, and comparing outputs. These operations were performed using Python (version 3.9.21) with Anaconda. This series of processes established an analytical framework that allows for the statistical and objective comparison and selection of various clustering methods. An overview of the clustering method development is shown in Figure 2.

#### 2.3.1. Data Pre-Processing

In the initial stage, foot IMU data from the GSTRIDE dataset during walking tests were used as input, specifically the *Y*-axis acceleration (progression direction), *Z*-axis acceleration (vertical direction), and *X*-axis angular velocity (sagittal plane rotation). The input data were segmented for each stride using the motion start and Foot Flat timings provided in the dataset. Subsequently, to reduce intra-subject variability, each of the three sensor signals (Y-accel, Z-accel, X-gyro) was processed as follows: all identified stride waveforms within the same subject were individually time-normalized to a uniform length. Then, a single mean waveform for each of the three axes was computed by averaging these normalized strides, resulting in three representative curves per subject.

#### 2.3.2. Feature Extraction

Principal Component Analysis (PCA) or t-distributed Stochastic Neighbor Embedding (t-SNE) was applied to map the averaged three-dimensional time series into a lower-dimensional space. The number of PCA components was determined based on a cumulative variance explained of 80%.

#### 2.3.3. Clustering Algorithm

Multidimensional Scaling (MDS) [25], using Dynamic Time Warping (DTW) as the distance function, was applied to the time series mapped into the low-dimensional space and to the basic data (age, height). DTW is an algorithm capable of calculating the distance between time series of different lengths [26], which allowed for the calculation of a dissimilarity matrix representing the distances between the representative strides of all subjects. The variables used for the distance matrix were configured in two separate patterns: one based solely on the dissimilarity of waveforms between subjects, and another combining waveform dissimilarity with basic data. Using this distance matrix, four types of clustering algorithms were tested: k-means++, k-medoids++, Fuzzy c-means, and hierarchical clustering. The number of clusters was validated for N = 3, 4, and 5.

#### 2.3.4. Comparing Outputs

Internal cluster validation was conducted to determine whether the clusters were well-separated from others, involving two steps: (1) a cutoff based on the effect size of key fall-related parameters, and (2) a quantitative comparison using cluster quality indices.

Optimizing clustering methods requires expert domain knowledge in addition to internal cluster indices. In this study, we introduced strict cutoffs based on the effect sizes of factors strongly associated with falls, as shown in previous studies, as a simpler alternative to expert weighting and knowledge intervention. For each condition, the Shapiro–Wilk test was performed for normality (α = 0.05). For conditions that meet the assumption of normality, a one-way ANOVA was conducted, and η^2^ was calculated as the effect size. For conditions where normality was not met, the Kruskal–Wallis test was used, and ε^2^ was calculated as the effect size. Subsequently, only conditions that exceeded a threshold (the minimum clinically meaningful effect size of 0.14 [27]) for four or more of the key fall-related parameters [28] (Cadence, Step Speed, Stride Length, Clearance, Swing, SPPB assessment, TUG) were used for the subsequent analysis.

The remaining conditions were evaluated through a comprehensive cluster quality analysis based on previous research [12], which included internal indices (Silhouette coefficient, Davies-Bouldin score, and Cluster Balance). Each index was max-min normalized, and their positive/negative directions were aligned before being linearly combined for the final evaluation.

### 2.4. Cluster Analysis

To quantify statistical differences, we compared the mean values of gait-related indices and clinical test results for each cluster and identified gait phenotypes based on these characteristics [14]. For post hoc comparisons, the Kruskal–Wallis test was applied, with the presence or absence of a fall history as the main axis of comparison.

### 2.5. Faller Classification Models

#### 2.5.1. Random Forest

All gait feature inputs were mean-imputed for any missing values and standardized (z-score normalization) before modeling. We trained a Random Forest classifier (scikit-learn RandomForestClassifier) to distinguish fallers from non-fallers, using a stratified K-fold cross-validation strategy (5 folds) to tune hyperparameters and evaluate performance. A grid search (scikit-learn version 1.6.1 GridSearchCV) was performed to optimize the number of trees (n_estimators = 100–300), maximum tree depth (including no limit and values 10–30), minimum samples required for a split (2, 5, 10), minimum samples per leaf (1, 2, 4), the feature subset strategy at splits (max_features = sqrt, log2, or all features), and bootstrap sampling usage. Model training in each fold used class-balancing weights to address the imbalanced fall class distribution. The F1-score (harmonic mean of precision and recall) was used as the scoring metric for model optimization, and the final model was refit on the entire dataset using the best hyperparameter combination (i.e., the one maximizing cross-validated F1). Classifier performance was evaluated by averaging the validation fold results, reporting the mean accuracy, precision (positive predictive value), recall (sensitivity), specificity, and F1-score.

#### 2.5.2. Extreme Gradient Boosting (XGBoost)

Preprocessing for the XGBoost model was identical, with mean imputation and feature standardization applied. We utilized an Extreme Gradient Boosting classifier (XGBClassifier from the XGBoost library) with a logistic objective for binary classification. Stratified 5-fold cross-validation and grid search were likewise employed to tune key parameters, including the number of boosting rounds (n_estimators = 100, 200, 300), maximum tree depth (3, 6, 9), learning rate (0.01, 0.1, 0.2), subsampling ratio (0.8, 0.9, 1.0), and column subsampling fraction per tree (0.8, 0.9, 1.0). As with the Random Forest, class imbalance was handled by applying class weights during training, and F1-score was the optimization target for the grid search. The best-found hyperparameters were used to train the final XGBoost model on the full dataset. We calculated the same suite of performance metrics (accuracy, precision, recall, specificity, F1-score) from the cross-validation results to assess the XGBoost classifier’s performance, using the confusion matrix of predictions to derive these measures.

#### 2.5.3. Decision Tree

We also developed a single Decision Tree classifier (DecisionTreeClassifier) using the same input preprocessing (mean imputation and standardization). Class weights (balanced) were applied to counteract the skewed class ratio, and we optimized the tree model’s complexity via a grid search with stratified 5-fold cross-validation. The hyperparameter search spanned the choice of split criterion (criterion: Gini or entropy), maximum tree depth (None or 5–30 levels), minimum samples for node split (2, 5, 10, 15, 20), minimum samples per leaf (1, 2, 4, 8, 10), maximum features considered at each split (all features, sqrt, or log2), and the splitting strategy (splitter: best vs. random). The grid search selected the combination yielding the highest F1-score on validation folds (using F1 as the scoring metric). We then refit the Decision Tree on the entire dataset with those optimal settings. The model’s effectiveness was evaluated in terms of accuracy, precision, recall (sensitivity), specificity, and F1-score, averaged over the cross-validation folds. These metrics, derived from confusion matrix results, characterized the Decision Tree’s classification performance.

#### 2.5.4. Artificial Neural Network (ANN)

For a non-ensemble approach, we implemented a feed-forward Artificial Neural Network using scikit-learn’s Multi-Layer Perceptron classifier (MLPClassifier). In preprocessing, missing feature values were filled with the median rather than the mean (a more robust choice for neural networks). We further performed outlier mitigation by identifying extreme values via the interquartile range (IQR) and clipping them to the acceptable range to reduce their influence. Because neural networks are sensitive to feature scaling, we applied a robust scaling (RobustScaler) which is resilient to outliers. Class imbalance was addressed by computing class weights (inverse-frequency weights) and incorporating them during training. The MLP architecture and training parameters were tuned with stratified 5-fold cross-validation and GridSearchCV. The hyperparameter grid included various network topologies (hidden_layer_sizes with 1–3 layers and 50–150 neurons per layer), activation functions (ReLU or tanh), L2 regularization strengths (alpha = 0.0001, 0.001, 0.01), initial learning rates (0.001, 0.01, 0.1), and the maximum number of training iterations (500, 1000, 1500). Early stopping was enabled to prevent overfitting, with the model reserving 10% of training data for validation and stopping if no improvement occurred in 10 epochs. We used the Adam optimizer for training. The best ANN configuration (as determined by highest CV F1-score) was retrained on the full dataset; an increased iteration limit was allowed in this final training to ensure convergence. The ANN’s performance was measured by cross-validated accuracy, precision, recall, specificity, and F1-score, analogous to the evaluation of the tree-based models.

#### 2.5.5. SHAP Interpretation

To interpret and compare feature importance across all classifiers, we employed SHapley Additive exPlanation (SHAP) analysis. For the tree-based models (Random Forest, XGBoost, and Decision Tree), we used the TreeSHAP algorithm via the SHAP TreeExplainer, which computes exact Shapley value contributions for each feature in tree ensembles. In our binary classification context, TreeExplainer returns separate SHAP value arrays for the negative and positive classes; we extracted the SHAP values corresponding to the positive class (falls) to explain each model’s predictions. These SHAP values indicate how much each feature contributes to pushing a given instance toward a fall or non-fall prediction. For the ANN, we utilized a kernel SHAP approach (KernelExplainer) since no analytical TreeSHAP exists for neural networks. We fit the KernelExplainer on a random subset of the data as a background distribution and defined the model prediction function to output the probability of the positive class. This yielded approximate SHAP values for the ANN, reflecting each feature’s influence on the predicted fall risk. To interpret results at the group level, we calculated the mean absolute SHAP value for each feature in each model as a measure of global feature importance. Comparing these values allowed us to identify which gait features were consistently influential in classifying fall risk across the different models. This TreeSHAP-based interpretability approach provided a unified, model-agnostic insight into feature importance, complementing the internal metrics of each classifier.

#### 2.5.6. Feature-Importance Consensus Ranking

To compare and integrate the feature importance among models, this study aggregated the rankings from multiple models using a weighted Borda scoring. The rank $r_{ij}$, assigned by each model j to each feature i, was linearly transformed into a Borda point $s_{ij}=n-r_{ij}+1$, where n is the number of features. These points were then multiplied by their respective model weights $w_{j}$ summed to calculate an overall score $S_{i}=\sum_{j=1}^{k}w_{j}\left(n-r_{ij}+1\right)$ for each feature. The model weights were determined by the accuracy of each model in the faller classification task. The final feature ranking was obtained by sorting these integrated scores in descending order.

### 2.6. Verifying the Impact of Screening Using a Single Indicator TUG

The accuracy of fall risk screening using the time taken for the TUG test as a single indicator was compared for the entire group of subjects and within each cluster. Conventionally, the TUG test is known as a simple indicator for screening high fall risk, with a common threshold of 13.50 s used to identify individuals at risk [10]. However, as mentioned earlier, this uniform cutoff has limitations in its sensitivity and specificity, and its universal application to the entire older adult population is problematic [7]. Therefore, in this study, we recalculated the optimal TUG time cutoff value for each cluster obtained through clustering and compared it with the conventional standard. Using the TUG scores and fall history data within each cluster, the threshold was determined as the point on the ROC curve that maximizes the Youden index (Sensitivity + Specificity − 1) [29]. The selection of the optimal cutoff based on the Youden index is a method for objectively determining the threshold that best balances sensitivity and specificity in a diagnostic test, and we applied this calculation within each cluster. For the calculated cluster-specific TUG thresholds, their sensitivity, specificity, and precision were determined from a confusion matrix and compared with the conventional uniform 13.50-s standard. We particularly focused on sensitivity (the proportion of fallers who can be screened without being missed) to evaluate the differences among clusters. The calculated thresholds should be considered exploratory reference values, not definitive clinical cutoffs, given the limited sample size in each cluster.

## 3. Results

### 3.1. Clustering Method

According to a comprehensive cluster analysis, the combination of PCA and k-medoids++ (with 4 clusters, using both signal waveform dissimilarity and basic data) ranked first among all conditions (Table 2). This combination demonstrated superiority in terms of cluster separation and balance and had the highest overall internal indices.

### 3.2. Cluster Characteristics

Table 3 shows the characteristics of the clusters constructed by the identified optimal clustering method. 146 participants were stratified into four different gait phenotypes. Each phenotype showed a unique profile in gait characteristics, physical function assessment, and fall prevalence. These quantitative profiles, shown in Table 3, were interpreted comprehensively and named High-cadence, Cautious, Intermediate and Robust as the names that best describe the characteristics of each group.

High-cadence group (n = 39) exhibited a distinctive gait signature. Their cadence was the highest of all groups (53.7 ± 4.55 steps/min), but their step speed was moderate (0.84 ± 0.24 m/s), meaning their stride length was shorter than that of the robust group. Their fall prevalence was high, at 56%. The term High-cadence directly reflects this most prominent gait characteristic. This gait pattern is suboptimal and can be interpreted as a compensatory strategy.

Cautious group (n = 34) was characterized by multiple parameters indicative of impaired physical function. Their fall prevalence, reaching 68%, was the highest of all groups. Their gait was the slowest (step speed: 0.67 ± 0.22 m/s) and their stride length was shortest (0.78 ± 0.19 m). In physical function tests, the SPPB score was lowest (7.00 ± 3.14), and the time required to complete the TUG test was longest (19.8 ± 9.84 s).

Intermediate group (n = 40) was intermediate between the Cautious and Robust groups in most indicators. Fall prevalence (48%), step speed (0.85 ± 0.28 m/s), and TUG time (13.0 ± 5.04 s) were all intermediate. This group was named Intermediate to reflect its position on the functional spectrum.

Robust group (n = 33) demonstrated the highest level of physical function. They had the lowest fall prevalence (27%), the fastest step speed (1.10 ± 0.31 m/s), and the longest stride length (1.14 ± 0.22 m). They also had the highest scores on functional tests (SPPB: 10.1 ± 1.85; TUG: 11.1 ± 4.19 s) and were named Robust to reflect their superior physical ability. However, the paradox of this group is that despite their high physical ability, approximately one-third of them experienced falls. Of note, the FES-I score (fear of falling) of those who fell was 14.3 ± 4.52, which was as high as that of other functionally inferior groups.

### 3.3. Classification Model Performance and Importance

The performance metrics for each model are shown in Table 4. In contrast to the overall accuracy of the four models being nearly equivalent, their suitability differed depending on the phenotype. For the High-cadence phenotype, ANN achieved the highest Accuracy (0.74) and F1-score (0.78). For the Cautious phenotype, RF achieved the highest Accuracy (0.89), Precision (0.90) and F1-score (0.93). For the Intermediate phenotype, RF was top in four metrics. For the Robust phenotype, Decision Tree had the highest Accuracy (0.91) and Precision (0.80), and ANN showed two highest metrics, but this phenotype also exhibited large variation across all models. The detailed model parameters are shown in Table A1. Note that these results were calculated to ensure the reliability of feature importance and as weights used to integrate importance rank, so no statistical tests were performed to show the superiority or inferiority of the models.

The rank of factors contributing to the risk of falls (top 10 features) for each model is shown in Figure 3. As shown in Figure 3, it should be noted that the feature importance of the Decision Tree was more concentrated on a few variables compared to the other models, resulting in many tied ranks for lower-importance features. The rank matrix and Weighted Borda overall scores for the top three features of each phenotype are presented in Table 5. The Borda score serves as an indicator of agreement among the models; a score exceeding 60 suggests a strong consensus on a feature’s importance. This analysis identified important features for each phenotype. After aggregating the ranks from the three tree-based models, Short FES-I was the top feature for the High-cadence and Robust groups, while Gait speed was the top features for the Cautious, and Push phase and 4 m Gait speed was the top feature for the Intermediate groups, revealing both similarities and differences among the phenotypes.

### 3.4. ROC Analysis of the Timed up and Go (TUG) Test

A comparison of the metrics for the conventionally used 13.50-s cutoff value and the customized threshold for each phenotype is shown in Table 6. As per Table 6, the classification performance improved for the Cautious and Intermediate groups. Across High-cadence group, Cautious group and Robust group, Recall increased, reducing the number of missed fallers; however, for the Robust group, the improvement in Recall was accompanied by a substantial decrease in Precision.

The widely cited 13.50-s cutoff originates from a study by Shumway-Cook et al. [10], which was established using a subject cohort where the TUG time distributions of faller and non-faller groups were exceptionally well-separated. Table 7 shows the characteristics of fallers in the previous study [10] and our study. The definition of fallers was stricter in the previous study [10], and the verification was conducted after elderly individuals with different functional abilities had already been clearly separated. In contrast, in this study, even after stratifying the subjects, overlap was observed in physical function performance. Particularly in the Cautious group and Intermediate group, the standard deviation of TUG time was large compared to the difference in mean values between the fallers and non-fallers groups, indicating a broad overlap in the distribution.

## 4. Discussion

### 4.1. Contribution of This Study

This study elucidates the heterogeneity in the gait of older adults using a data-driven approach, presenting a novel perspective for personalized fall risk assessment. The innovative finding of our study is the clear demonstration that the key factors associated with a history of falls differ across these phenotypes, and consequently, the optimal screening threshold for the widely used Timed Up and Go (TUG) test also varies for each phenotype. These results corroborate the recent consensus that a one-size-fits-all approach is insufficient for fall risk stratification in older adults [7,8,9]. This underscores the necessity for personalized preventive strategies tailored to specific gait phenotypes, moving beyond a uniform approach to fall risk management.

### 4.2. Different Key Factors Across Phenotypes

A key finding of this study is that the most significant features for fall risk stratification differ across gait phenotypes. Our analysis of feature importance suggests that the mechanisms leading to falls in older adults are not uniform but rather follow at least two primary pathways. The first pathway, declined physiological reserve, is prominent in the Cautious and Intermediate phenotypes, which are characterized by relatively low physical function. In this pattern, a decline in physical capacity itself, typified by slowed or less stable gait, serves as direct trigger for falls. For example, the mean 4 m gait speed of fallers in Cautious group was 0.59 ± 0.14 m/s, significantly below the frailty criteria. This explains why indicators of reduced gait ability (e.g., reduced gait speed) were one of the most important factors in classifying fallers in these phenotypes (Borda score: 83.37 for Cautious group, 68.25 for Intermediate group), since these reflect overall physical ability [30,31]. The second pathway, impaired psychological control, is dominant in the High-cadence and Robust phenotypes with relatively high physical function. For these individuals, falls are not an event of physical limitation but rather one where psychological factor, namely Fear of Falling (FoF), inappropriately interfere with the motor control [32,33]. Despite sufficient physical capacity, this interference results in attentional allocation failures and inappropriate motor strategies, which is why the Short FES-I (a scale for FoF) was one of the most critical factors for this group.

The Cautious group corresponds to a typical frail group, characterized by the slowest average gait speed, the shortest step length, the lowest Short Physical Performance Battery (SPPB) score, and the highest fall prevalence at 68%. Our classification models identified 4 m Gait speed as the most powerful predictor of fall history for this group (Borda Score: 83.37). This finding is consistent with a large body of evidence establishing gait speed as a “sixth vital sign” that sensitively reflects overall health status, functional capacity, and physiological reserve [30,31,34]. Crucially, the second most important feature was the percentage of the gait cycle spent in the Load phase (Borda Score: 64.65). The loading response is the initial period of double support, critical for shock absorption and stable weight transfer. An increased Load percentage, coupled with a globally reduced gait speed, signifies a deliberate strategy to maximize stability at the expense of forward progression and efficiency. This pattern reflects compensation for diminished balance and muscle weakness, where individuals prolong the time with both feet on the ground to ensure a stable base of support before committing to the more challenging single-limb stance phase [35]. Thus, the fall mechanism in this phenotype is not attributable to a single isolated deficit but rather to a global failure of the locomotor system. The mechanism of falls in this phenotype can thus be interpreted as a state where the capacity of multiple physiological systems—such as the neuromuscular, cardiovascular, and sensory systems—becomes insufficient to meet the physical demands required for walking and maintaining posture [36]. Furthermore, in this group, FES-I scores were uniformly high for both fallers and non-fallers, with no significant difference between them. This suggests that FoF is a rational response based on an accurate perception of one’s own declining physical abilities, and is likely a consequence of physical decline rather than the primary determinant of the gait pattern [32]. From the above, the cautious group adopted a cautious gait pattern, intentionally slowing their walking and lengthening the load phase, to compensate for their decreased balance and muscle strength (i.e., frailty). However, this compensatory strategy does not even meet the minimum walking ability required in daily life, and it can be interpreted as a breakdown of the entire system, leading to falls. This is not a fall due to an inability to respond to a specific disturbance (trip/slip), but rather a failure to maintain voluntary gait itself.

The Intermediate group is positioned as a transitional group, having higher physical function than the Cautious group but still facing a fall risk. Strikingly, our models revealed that the most critical feature for this group was not a global measure of decline, but a specific biomechanical deficit: the percentage of the gait cycle in the Push phase (Borda Score: 69.00), followed closely by 4 m Gait speed (Borda Score: 68.25). The Push phase, or pre-swing, is where the ankle plantarflexion muscles generate most of the propulsive power required to drive the body forward into the next step [37,38]. A decline in ankle push-off power is a cardinal sign of aging gait, and interventions targeting this mechanism have shown promise for improving locomotor function. The emergence of Push ratio as the top predictor in this functionally intermediate group suggests a hierarchical pattern of decline. It implies that the failure of the primary propulsive engine at the ankle may be one of the earliest critical biomechanical failures that precedes the more global deterioration observed in the Cautious phenotype. This phenotype, therefore, should not be viewed merely as an intermediate-risk group, but as representing a critical stage where a specific, addressable dysfunction in the motor control system has begun to manifest. When propulsive force decreases, the leg does not swing forward sufficiently during the swing phase. As a result, the leg does not rise sufficiently and gets caught on the ground, or the stride becomes too short, causing the body’s center of gravity to deviate from the base of support and the body weight to be unable to support itself on the next step, which is thought to be the mechanism behind the fall. This differs from the overall speed reduction seen in the Cautious group and can be interpreted as a specific malfunction of the propulsion system.

One of the notable findings of this study is that Short FES-I was the primary important factor for classifying fall risk in the High-cadence and Robust phenotypes, both of which have relatively high physical function. FoF is not only a consequence of falls but also a powerful fall risk factor in its own right, known to exist even in individuals who have never fallen [32,33]. Our results show that this single psychological driver can lead to two distinct, maladaptive biomechanical expressions. For the Robust phenotype, which exhibited the best physical performance, Short FES-I (Borda Score: 67.90) was coupled with Stride time (STD) (gait variability) (Borda Score: 70.53) as top predictors. This combination suggests a mechanism of psychologically induced breakdown of gait automaticity. Anxious but otherwise healthy older adults do not necessarily adopt a slow, cautious gait. Instead, their fear appears to interfere with the subconscious, automatic control of walking [39,40]. This conscious control is less efficient and disrupts the natural rhythm of gait, leading to increased stride-to-stride variability [41]. This increased variability is a powerful and independent predictor of future falls, particularly in faster-walking older adults, which describes the Robust group. Previous research has shown that increased gait variability is directly linked to a decreased ability to recover from unexpected disturbances (i.e., trips and slips) [42,43]. In contrast, the High-cadence phenotype represents a different response to the same psychological driver. Here, Short FES-I (Borda Score: 62.10) was the top predictor, but the biomechanical signature was not increased variability but rather a distinct gait pattern characterized by high Cadence and short Stride time. This pattern is a well-known compensatory strategy used to increase stability in response to perceived instability or FoF. By taking more frequent, shorter steps, individuals minimize the time spent in the less stable single-limb support phase and maximize the duration of double support. While this may provide a subjective sense of security, this adapted gait pattern is less efficient and less adaptable to environmental challenges, such as clearing obstacles or recovering from a large perturbation, thus paradoxically increasing their fall risk in real-world settings [32]. These two phenotypes compellingly illustrate that in physically capable older adults, the pathway to falls can be primarily psychological.

### 4.3. Validity of Phenotype-Specific TUG Thresholds

While the Timed Up and Go (TUG) test is a fundamental tool for assessing motor function in older adults, meta-analyses have pointed out the limitations of using a single cutoff value, such as 13.50 s, for fall prediction validity, particularly its low sensitivity [7,8,9]. Although it has been suggested that the heterogeneity of the target population underlies this limitation [11,44], the significance of our study lies in being the first to demonstrate that this heterogeneity can be systematically explained by gait phenotypes.

Previous study [10] used a stricter definition of fallers and conducted its analysis only after clearly distinguishing older adults with differing functional abilities. In their data, the TUG time for the non-faller group ranged from 6.4 to 12.6 s, with minimal overlap with the faller group’s range of 10.3 to 39.2 s. Such a clear separation is rare in broader, more diverse populations of community-dwelling older adults. Therefore, one of the core challenges in fall risk assessment is the considerable overlap in physical function performance between individuals with and without a fall history. Our study revealed that this overlap persists even after stratifying subjects into distinct gait phenotypes. Particularly in the Cautious and Intermediate phenotypes, the standard deviations of TUG times were large relative to the difference in mean values between the faller and non-faller groups, resulting in extensive overlap of their distributions. This inherent overlap suggests that classifying fallers using the single metric of TUG time has fundamental limitations in its potential accuracy, regardless of any universal single threshold employed.

The customization of thresholds in this study is a direct response to this challenge. The fact that the customized TUG thresholds varied from 11.95 to 14.00 s depending on the phenotype reflects the differing baseline motor performance of each group. The clinical utility derived from this personalized strategy was not uniform across all phenotypes. For instance, in the Cautious group, adjusting the threshold to 12.79 s improved both sensitivity and the F1 score, indicating a clear diagnostic benefit for this high-risk population. On the other hand, in the Robust group, lowering the threshold to 12.00 s improved sensitivity but at the cost of a significant trade-off, namely, a substantial decrease in precision. This outcome suggests that for this specific phenotype, the utility of the TUG test as a standalone screening tool may be limited, which aligns with another finding of our study: psychological factors (fear of falling) were more critical than physical function in predicting fall risk for this group. Other previous studies [11,45,46] have also pointed out that the TUG test alone has low classification power among relatively healthy and high-functioning older adults. The results of this study align with previous research in showing that the TUG has low classification power in highly functional older adults. Furthermore, it is valuable in suggesting that psychological interventions, rather than physical functional ones, may be effective for such healthy groups. The methodology of this study is linked to more specific individualized approaches than conventional subgrouping.

In conclusion, while phenotype-specific threshold setting did not yield perfect classification due to the inherent data overlap, the strategy itself proved to be more valid and clinically more refined than using a uniform standard. The value of this research lies not in providing a definitive new cutoff value, but in demonstrating that a personalized, phenotype-aware approach is indispensable. It represents a paradigm shift from the goal of finding a single correct threshold to tailoring the diagnostic properties of a screening test to the characteristics and clinical needs of the target subgroup. This substantiates the need to move away from a one-size-fits-all approach in fall risk management.

### 4.4. Limitations of This Study

This study has several limitations. The biggest limitation of this study is its reliance on only a single cohort (the GSTRIDE database). The robustness of this phenotype classification to data collected at different facilities, with different IMU sensors, and using different protocols remains to be verified. The lack of large-scale publicly available datasets with fall history and IMU waveform data other than the GSTRIDE database makes validation difficult at this time. Furthermore, its cross-sectional design precludes the inference of causality. Although we found associations between features and fall history, we cannot conclude that these factors cause falls. The history of falls itself may alter gait and increase FoF. Additionally, fall history was based on self-report over the past 12 months and may be subject to recall bias. Furthermore, while the overall sample size was sufficient for clustering and initial analysis, the subgroup sample sizes for comparing fallers and non-fallers within each phenotype were relatively small, which may have limited the statistical power in some secondary analyses.

## 5. Conclusions

This study elucidated the heterogeneity of gait in older adults through a data-driven approach, objectively identifying four distinct gait phenotypes: High-cadence, Cautious, Intermediate, and Robust.

First, the primary risk factors for falls differed distinctly among the phenotypes. For Cautious and Intermediate phenotypes, characterized by lower physical function, fall risk is primarily driven by a decline in physiological reserve. This decline manifested itself in both Cautious and Intermediate groups as a decrease in overall gait speed, while in the latter type it manifested as a change in ankle push phase. In contrast, for the physically capable High-cadence and Robust groups, fall risk was linked to impairments in psychological and attentional control. This pathway is driven by Fear of Falling (Short FES-I) in High-cadence group, leading to a compensatory high-cadence strategy. In the Robust group, this psychological factor, along with fear of falling, was most significantly manifested in increased gait variability (Stride time STD), suggesting a breakdown in automated motor control.

Second, the customized screening threshold for the TUG test also varied significantly according to the phenotype, ranging from 11.95 to 14.00 s, which empirically demonstrates the limitations of a one-size-fits-all standard such as the conventional 13.50 s cutoff.

These findings strongly suggest that fall mechanisms are phenotype-dependent, underscoring that a personalized assessment strategy—one that considers the unique interplay of physiological and psychological factors within each subgroup—is indispensable for fall risk assessment in older adults.

## Figures and Tables

**Figure 1 sensors-25-07503-f001:**
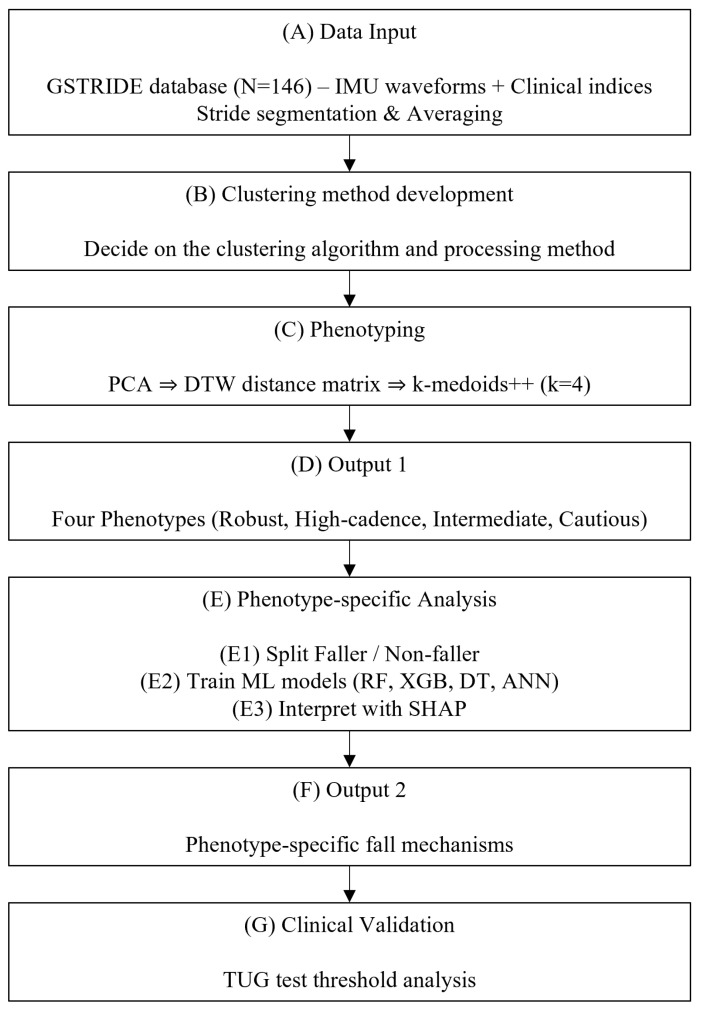
Schematic diagram.

**Figure 2 sensors-25-07503-f002:**
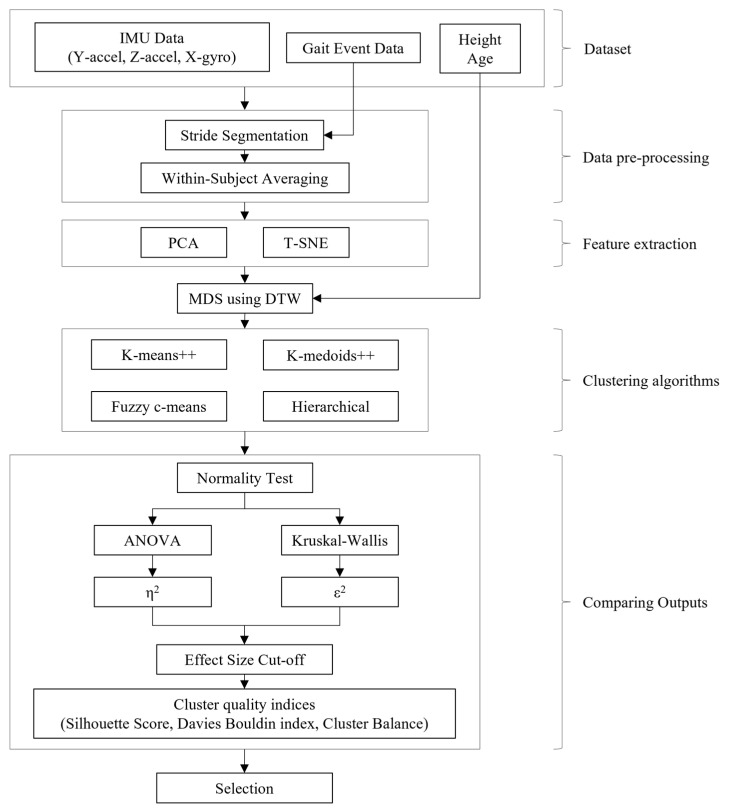
Clustering method development flow.

**Figure 3 sensors-25-07503-f003:**
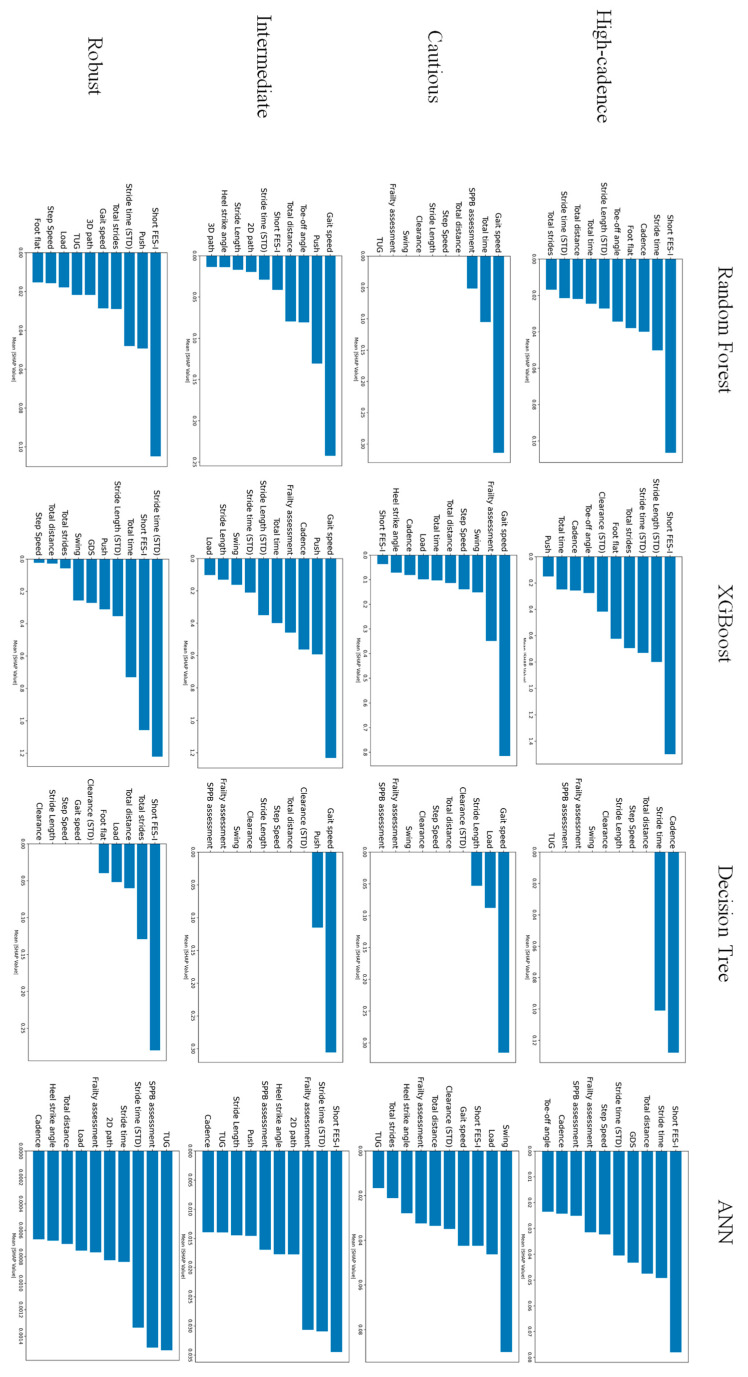
Comparison of top features by feature importance calculated for each model.

**Table 1 sensors-25-07503-t001:** Features. Except for features containing STD or Total in their names, walking-related parameters are averages for the entire walking test. A figure of gait phases and sub-phases is shown in Figure A1. GCT: Gait Cycle Time.

Feature	Description
Cadence [strides/min]	Number of strides per minute according to the time spent in walking the stride *n*.
Step Speed [m/s]	Average linear speed during the stride *n*.
Stride Length [m]	Length between the initial and end points of the stride *n*. It is calculated as the increment in the XY position (horizontal plane) between the initial and final points of the stride.
Clearance [m]	Elevation of the foot during the swing phase of the stride *n*. It is calculated as the maximum height that the foot reaches during swing phase.
Total Distance [m]	The total distance covered during a long free walk.
Total Time [s]	The total time spent in the free walk.
Total Strides	The total number of steps in the free walk.
Swing [% GCT]	Time percentage of the stride *n* during which the foot is in the Swing phase. It is calculated as the elapsed time between Toe-Off and Heel-Strike events.
Load [% GCT]	Time Percentage of the stride *n* during which the foot is in the Load phase. It is calculated as the elapsed time between the Heel-Strike event and Toe-Strike (TS).
Foot Flat [% GCT]	Percentage of the stride *n* during which the foot is in the Foot Flat phase, i.e., the foot is completely in contact with the ground, corresponds with central part of the stance phase. It is calculated as the elapsed time between Toe-Strike (TS) and Heel-Off (HO) events detected.
Push [% GCT]	Percentage of the stride *n* during which the foot is in the Push phase. It is calculated as the time elapsed from the Heel-Off (HO) until the Toe-Off occurs.
Stride Time [s]	Time spent to walk the stride n. It is calculated by the time elapsed between the beginning and the end events of a stride.
Toe-off Angle [deg]	Pitch angle of the foot in the Toe Off moment of the stride *n*.
Heel Strike Angle [deg]	Pitch angle of the foot in the Toe Off moment of the stride *n*.
Stride Time STD [s]	The standard deviation of stride time in trials.
Stride Length STD [m]	The standard deviation of stride time in trials.
Clearance STD [m]	The standard deviation of clearance in trials.
3D Path [m]	Length of the 3D trajectory of the stride *n*. It is calculated as the cumulative three-dimensional displacement (XYZ) made during the stride.
2D Path [m]	Length of the projection of the 3D trajectory of the stride *n* into the horizontal plane. It is calculated as the cumulative horizontal displacement (XY) made during the stride.
GDS	Global Deterioration Scale [21].
Frailty Assessment	Fried’s frailty scale [22].
Short FES-I	The Short Falls Efficacy Scale-International [23].
SPPB Assessment	Short Physical Performance Battery [24].
4 m Gait Speed [m/s]	The walking speed in 4-meter walking test.
TUG time [s]	Time taken for the Timed Up and Go test.

**Table 2 sensors-25-07503-t002:** Details of the selected clustering method.

Feature Extraction	PCA
Clustering algorithm	k-medoids++
Number of clusters	4
Used features	Raw gait signals + body measurement data (height, age)
Silhouette coefficient	0.198
Davies Bouldin score	1.08
Cluster Balance ^1^	7

^1^ Difference between maximum cluster size and minimum cluster size.

**Table 3 sensors-25-07503-t003:** Characteristics of phenotypes. Mean ± standard deviation within clusters.

Parameters		High-Cadence (n = 39)	Cautious (n = 34)	Intermediate (n = 40)	Robust (n = 33)
Fall prevalence (%)	−	56	68	48	27
Step speed (m/s)	All	0.84 ± 0.24	0.67 ± 0.22	0.85 ± 0.28	1.10 ± 0.31
	NF	0.94 ± 0.23 *	0.83 ± 0.25 *	0.92 ± 0.25	1.20 ± 0.28 *
	F	0.76 ± 0.21	0.59 ± 0.14	0.77 ± 0.29	0.83 ± 0.17
Stride length (m)	ALL	0.87 ± 0.23	0.78 ± 0.19	0.95 ± 0.25	1.14 ± 0.22
	NF	0.93 ± 0.20	0.90 ± 0.19 *	1.03 ± 0.25 *	1.21 ± 0.20 *
	F	0.82 ± 0.23	0.73 ± 0.16	0.85 ± 0.21	0.95 ± 0.16
Cadence (steps/min)	ALL	53.7 ± 4.55	46.5 ± 5.86	48.3 ± 6.34	51.9 ± 6.06
	NF	55.7 ± 3.61 *	50.5 ± 5.87 *	48.2 ± 5.09	53.4 ± 5.79 *
	F	52.1 ± 4.59	44.6 ± 4.81	48.5 ± 7.48	47.7 ± 4.62
SPPB assessment	ALL	8.49 ± 2.43	7.00 ± 3.14	9.15 ± 2.23	10.1 ± 1.85
	NF	9.06 ± 2.34	8.91 ± 3.58 *	10.1 ± 1.85 *	10.4 ± 1.91
	F	8.05 ± 2.40	6.09 ± 2.43	8.11 ± 2.15	9.44 ± 1.50
TUG time (s)	ALL	15.9 ± 5.58	19.8 ± 9.84	13.0 ± 5.04	11.1 ± 4.19
	NF	14.2 ± 3.81	16.2 ± 12.6 *	11.2 ± 4.24 *	9.44 ± 2.63 *
	F	17.2 ± 6.36	21.7 ± 7.36	15.4 ± 5.01	15.6 ± 4.33
FES-I	ALL	11.2 ± 4.57	11.4 ± 6.17	9.86 ± 3.99	9.58 ± 4.01
	NF	8.50 ± 2.50 *	9.88 ± 4.34	8.84 ± 3.63 *	7.64 ± 1.11 *
	F	13.5 ± 4.68	12.1 ± 6.68	11.0 ± 4.06	14.3 ± 4.52

* shows *p* < 0.05 in statistical tests conducted between NF and F. ALL: All subjects. NF: Non-Faller. F: Faller.

**Table 4 sensors-25-07503-t004:** Model performance metrics. Mean (95% Confidence Interval) of k-fold CV.

Phenotypes		Random Forest	XGBoost	Decision Tree	ANN
High-cadence	Accuracy	0.74 (0.51, 0.96)	0.71 (0.52, 0.91)	0.61 (0.47, 0.76)	0.74 (0.58, 0.89)
	Precision	0.79 (0.56, 1.00)	0.73 (0.51, 0.95)	0.60 (0.48, 0.72)	0.72 (0.59, 0.85)
	Recall	0.80 (0.57, 1.00)	0.80 (0.57, 1.00)	1.00 (1.00, 1.00)	0.85 (0.70, 1.00)
	F1-score	0.77 (0.60, 0.94)	0.75 (0.57, 0.94)	0.75 (0.65, 0.84)	0.78 (0.64, 0.92)
Cautious	Accuracy	0.89 (0.71, 1.00)	0.86 (0.70, 1.00)	0.83 (0.62, 1.00)	0.80 (0.71, 0.88)
	Precision	0.90 (0.75, 1.00)	0.84 (0.67, 1.00)	0.82 (0.62, 1.00)	0.84 (0.67, 1.00)
	Recall	0.96 (0.86, 1.00)	1.00 (1.00, 1.00)	1.00 (1.00, 1.00)	0.91 (0.77, 1.00)
	F1-score	0.93 (0.81, 1.00)	0.91 (0.81, 1.00)	0.89 (0.76, 1.00)	0.86 (0.81, 0.91)
Intermediate	Accuracy	0.78 (0.57, 0.98)	0.73 (0.46, 0.99)	0.75 (0.49, 1.00)	0.75 (0.53, 0.97)
	Precision	0.82 (0.54, 1.00)	0.81 (0.50, 1.00)	0.80 (0.49, 1.00)	0.75 (0.49, 1.00)
	Recall	0.80 (0.57, 1.00)	0.75 (0.41, 1.00)	0.75 (0.47, 1.00)	0.80 (0.57, 1.00)
	F1-score	0.77 (0.60, 0.95)	0.71 (0.44, 0.98)	0.74 (0.51, 0.98)	0.76 (0.56, 0.95)
Robust	Accuracy	0.84 (0.71, 0.97)	0.90 (0.81, 1.00)	0.91 (0.82, 1.00)	0.79 (0.57, 1.00)
	Precision	0.73 (0.25, 1.00)	0.73 (0.25, 1.00)	0.80 (0.30, 1.00)	0.65 (0.28, 1.00)
	Recall	0.60 (0.14, 1.00)	0.70 (0.20, 1.00)	0.60 (0.14, 1.00)	1.00 (1.00, 1.00)
	F1-score	0.63 (0.21, 1.00)	0.69 (0.24, 1.00)	0.67 (0.21, 1.00)	0.75 (0.48, 1.00)

**Table 5 sensors-25-07503-t005:** Rank matrix. Only the top three features in Borda overall score for each phenotype are listed. The machine learning model column shows the ranking of feature importance for each model. A Borda overall score exceeding 60 indicates good agreement between models and that this feature is an important feature.

Phenotypes	Features	Random Forest	XGBoost	Decision Tree	ANN	Borda Overall Score
High-cadence	Short FES-I	1	1	14	1	62.10
	Cadence	3	8	1	10	56.96
	Stride time	2	19	2	3	54.40
Cautious	4 m Gait speed	1	1	1	2	83.37
	Load	14.5	7	2	3	64.65
	Swing	14.5	3	14.5	1	59.31
Intermediate	Push	2	2	2	6	69.00
	4 m Gait speed	1	1	1	10	68.25
	Stride time (STD)	6	7	14	3	55.53
Robust	Stride time (STD)	3	1	15.5	2	70.53
	Total strides	4	8	2	9	70.10
	Short FES-I	1	2	1	23	67.90

**Table 6 sensors-25-07503-t006:** TUG classification metrics. In the Customized column, the Threshold row shows the customized threshold [s], and the Performance Index rows show the faller identification results for those thresholds.

Phenotypes		13.50 s Cutoff	Customized
High-cadence	Threshold [s]	13.50	11.95
	Precision	0.67	0.65
	Recall	0.7	0.75
	F1 score	0.68	0.70
	Accuracy	0.64	0.64
Cautious	Threshold [s]	13.50	12.79
	Precision	0.83	0.84
	Recall	0.79	0.84
	F1 score	0.81	0.84
	Accuracy	0.76	0.79
Intermediate	Threshold [s]	13.50	14.00
	Precision	0.67	0.75
	Recall	0.67	0.60
	F1 score	0.67	0.67
	Accuracy	0.71	0.74
Robust	Threshold [s]	13.50	12.00
	Precision	0.75	0.58
	Recall	0.67	0.78
	F1 score	0.71	0.67
	Accuracy	0.85	0.79

**Table 7 sensors-25-07503-t007:** Comparison of characteristics of fallers. NF: Non-Faller. F: Faller.

	Shumway-Cook et al. [10]	Our Study Using the GSTRIDE Database [15]	
Definition of Faller	Elderly people who have fallen two or more times within the past six months	Elderly people who have fallen at least once within the past year	
TUG time average (s)	NF: 8.4 ± 1.7 F: 22.2 ± 9.3	High-cadence	NF: 14.2 ± 3.81 F: 17.2 ± 6.36
		Cautious	NF: 16.2 ± 12.6 F: 21.7 ± 7.36
		Intermediate	NF: 11.2 ± 4.24 F: 15.4 ± 5.01
		Robust	NF: 9.44 ± 2.63 F: 15.6 ± 4.33
TUG time range (s)	NF: 6.4–12.6 F: 10.3–39.2	High-cadence	NF: 10.3–22.2 F: 9.59–35.0
		Intermediate	NF: 5.26–50.0 F: 9.78–36.0
		Cautious	NF: 5.79–23.5 F: 7.74–25.9
		Robust	NF: 6.00–16.5 F: 9.30–23.0

## Data Availability

This database containing all the IMU recordings, estimated parameters in Excel format, and the Python code to process the IMU signals are available in a Zenodo repository [47].

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
