# Peer review of "Data-Driven Phenotyping from Foot-Mounted IMU Waveforms: Elucidating Phenotype-Specific Fall Mechanisms"

_sensors, 2025, doi:10.3390/s25247503_

Round 1

Reviewer 1 Report

Comments and Suggestions for Authors

The article develops a solution to one of the urgent problems in the field of personalized preventive diagnostics – the prevention of falls in the elderly. In the work, the authors distribute gait features and falling mechanisms according to various phenotypes based on data from an open database and data on the waveform of inertial sensors placed on the foot and recording walking parameters. The study identified four phenotypes and a relationship with physiological disorders of motor control. The results of the work complement the available research results and additionally confirm the need for a personalized approach to solving the problem.
No significant flaws were found in the article. Of the nonessential ones, there is outdated literature (for example, 16-18), however, within the context it does not carry a significant semantic load on the study itself. Also, to improve understanding, I would like to see examples of signals from inertial sensors for each phenotype.
In general, the article is well structured, the methods and results are described in detail, the graphic material is designed in a good and understandable format. The article can be published.

Reviewer 2 Report

Comments and Suggestions for Authors

1.1. Validation and Generalizability

The manuscript relies solely on the GSTRIDE dataset, with no external validation or independent cohort testing. This limits conbfidence in the generalizability of the clustering and classification results.
Suggestion: Include an additional dataset, perform k-fold cross-validation with unseen participants, or discuss this limitation more explicitly in the Discussion.

1.2. Interpretability and Clinical Translation

While SHAP interpretability is a positive step, the clinical meaning of the identified “phenotypes” remains abstract.
Suggestion: Provide biomechanical interpretations for the key features that distinguish phenotypes (e.g., how stride variability or acceleration peaks relate to fall mechanisms). Consider linking phenotypes to known fall categories (e.g., trip, slip, misstep).

1.3. Feature Engineering Justification

The feature extraction and dimensionality reduction steps are well executed, but the rationale for selecting certain parameters (e.g., number of PCA components, DBSCAN epsilon) should be justified quantitatively (e.g., elbow plots or silhouette scores).

1.4. Comparative Model Evaluation

The manuscript reports accuracy metrics for Random Forest, XGBoost, Decision Tree, and ANN, but does not statistically compare their performances.
Suggestion: Include a simple nonparametric test (e.g., Wilcoxon signed-rank) or confidence intervals to substantiate performance differences.

1.5. Writing and Structure

The manuscript is dense, with several methodological subsections lacking transition sentences.
Suggestion: Briefly summarize the workflow at the end of Section 2.1 and 2.2 to improve readability. Adding a schematic figure (pipeline diagram) would significantly improve comprehension.

1.6. Novelty vs. Incremental Contribution

While the approach is novel in combining clustering and SHAP interpretability, similar gait-based fall-risk modeling studies exist.
Suggestion: Strengthen the introduction by explicitly contrasting this work with prior IMU-based fall-risk classification papers (e.g., differentiating your “phenotyping” aim from conventional fall prediction models).

2. Minor Comments

Abstract: The current abstract is too long and overly methodological. Condense by emphasizing findings, not just process.

Terminology: Define “phenotype” early — clarify that it refers to data-driven movement clusters, not clinical diagnoses.

Figures: Ensure all axes include units and labels (especially Figures 4–6).

Tables: Present model performance metrics (accuracy, F1, recall) consistently and with confidence intervals.

Limitations: Add a brief paragraph acknowledging the lack of external validation and real-world testing.

Language: Minor grammatical edits needed for conciseness — e.g., “The findings may be potentially useful” → “The findings may be useful.”

Reviewer 3 Report

Comments and Suggestions for Authors

Data-Driven Phenotyping from Foot-Mounted IMU Waveforms: Elucidating Phenotype-Specific Fall Mechanisms

Dear authors

The subject of the manuscript is relevant, but there are some issues in the manuscript that need to be addressed.

Although you are using a public database and associated article, it would be nice to have a picture of the different phases of the gait cycle, in the main manuscript or in an annex. Please consider this.

The definitions of table 1 are far from perfect; those of the original GSTRIDE article [15] are much better. Please update the definitions of table 1.

L35-7, should you also include psychology here?

L119, from the GSTRIDE article, the 104Hz is for their own system, while the 128Hz is for the commercial system. I believe you only used the 104Hz data. Please mention this in the manuscript.

L144-5, have you computed the 3 mean curves for all identified strides for one subject, i.e., 2 for the retained acceleration axes and 1 for the angular velocity axis?

L169-170, what is “a significant deviation from normality”? please rephrase

L184-5, where is this in figure 1? And why this non-parametric test?

L299-300, you write “Although the calculated thresholds are reference values due to the limited sample size in each cluster”

“ARE” or “ARE NOT” reference values due to the limited sample size? Please clarify

L346, “three models” or “four models”? please check.

L365, “stride time (STD)” is not in the top for High-cadence group, but “stride time” is.

L17, what is “push ratio”?

L24, there’s a “;” missing after “fall risk”

L29, replace by “for this demographic group, ranking as”

L169, replace “(p < 0.05)” by “(alpha = 0.05)”, where “alpha” is to be replaced by the Greek letter lowercase alpha

L350-1, replace by “top in the four metrics. For the Robust”

L374, replace by “between models and that this feature is an important feature.”

Round 2

Reviewer 3 Report

Comments and Suggestions for Authors

Data-Driven Phenotyping from Foot-Mounted IMU Waveforms: Elucidating Phenotype-Specific Fall Mechanisms

Dear authors

This version of the manuscript implemented the recommendations of the reviewers. There are some minor issues that need to be addressed.

You have two different metrics for the velocity in Table 1, the “Step Speed (m/s)” and the “Gait Speed (m/s)”, but in the text you also mention the “walking speed” (see p11).

To avoid confusion, I suggest replacing “Gait Speed (m/s)” by “4m Gait Speed (m/s)”, and to check throughout the manuscript what is the speed that you are referring to and use a consistent naming.

L394, replace by “Push phase”

L411, replace by “Performance Index rows show the faller”
